# The Plastidial Glyceraldehyde-3-Phosphate Dehydrogenase Is Critical for Abiotic Stress Response in Wheat

**DOI:** 10.3390/ijms20051104

**Published:** 2019-03-04

**Authors:** Xixi Li, Wenjie Wei, Fangfang Li, Lin Zhang, Xia Deng, Ying Liu, Shushen Yang

**Affiliations:** College of Life Sciences, Northwest A&F University, Yangling 712100, Shaanxi, China; lixixi2018wlf@163.com (X.L.); wenjiewei2011@163.com (W.W.); liffnwafu@163.com (F.L.); Lin798335901@163.com (L.Z.); 18209272902@163.com (X.D.)

**Keywords:** TaGAPCp1, yeast two-hybrid system (Y2H), BiFC, abscisic acid (ABA), hydrogen peroxide (H_2_O_2_)

## Abstract

Plastidial glyceraldehyde-3-phosphate dehydrogenase (GAPDH, GAPCp) are ubiquitous proteins that play pivotal roles in plant metabolism and are involved in stress response. However, the mechanism of GAPCp’s function in plant stress resistance process remains unclear. Here we isolated, identified, and characterized the *TaGAPCp1* gene from Chinese Spring wheat for further investigation. Subcellular localization assay indicated that the TaGAPCp1 protein was localized in the plastid of tobacco (*Nicotiana tobacum*) protoplast. In addition, quantitative real-time PCR (qRT-PCR) unraveled that the expression of TaGAPCp1 (GenBank: MF477938.1) was evidently induced by osmotic stress and abscisic acid (ABA). This experiment also screened its interaction protein, cytochrome b6-f complex iron sulfite subunit (Cyt b6f), from the wheat cDNA library using TaGAPCp1 protein as a bait via the yeast two-hybrid system (Y2H) and the interaction between Cyt b6f and TaGAPCp1 was verified by bimolecular fluorescence complementation assay (BiFC). Moreover, H_2_O_2_ could also be used as a signal molecule to participate in the process of Cyt b6f response to abiotic stress. Subsequently, we found that the chlorophyll content in OE-*TaGAPCp1* plants was significantly higher than that in wild type (WT) plants. In conclusion, our data revealed that TaGAPCp1 plays an important role in abiotic stress response in wheat and this stress resistance process may be completed by H_2_O_2_-mediated ABA signaling pathway.

## 1. Introduction

Wheat (*Triticum aestivum* L.) is one of the most important food crops in the world. However, crop yields have fallen sharply due to climate change and widespread drought [1,2]. Glyceraldehyde-3-phosphate dehydrogenase (GAPDH) is a key enzyme in the glycolytic pathway [3]. GAPDH was once considered a simple “housekeeping gene”. Therefore, it is often used as a reference for gene expression and protein research [4]. 

However, GAPDH has recently been shown to play a vital role in many cellular processes, like energy production, DNA repair, transcriptional regulation, sugar and amino acid balance, embryo development, viable pollen development, root growth and abscisic acid (ABA) signal transduction [5,6], except for glycolysis [7,8,9]. Studies have shown that *GAPDH* is divided into four subfamilies of *GAPA/B*, *GAPC*, *GAPCp* and *GAPN* in plant cells [10,11,12], renamed subfamily I, subfamily II, subfamily III and subfamily IV, separately. In Arabidopsis, phosphorylated *GAPDH* contains two *gapA*, two *gapC* and two *gapCp* genes, with only a single *gapB* [6,13]. According to previous studies, there were 13 *GAPDH* genes in wheat, including five *gapCp* genes [14]. GAPCp was found in the angiosperms and originated in early chloroplast evolution through duplication of the cytosolic *gapC* gene [15]. 

In spite of the low gene expression level of GAPCps in contrast with cytosolic GAPDHs, GAPCp still participates in glycolytic energy production and metabolic regulation in non-green plastids [12,16,17]. GAPCp also is an important metabolic connector of carbon and nitrogen metabolism through the phosphorylated pathway of serine biosynthesis. Additionally, studies have shown that GAPCp1 and GAPCp2 are located in the chloroplast in Arabidopsis [6]. Mutations in the *GAPCp* gene cause metabolic abnormalities in the triose phosphate transporter (TPT) [18]. Transcriptome and metabolomic analysis have shown that the lack of GAPCp in plants can disrupt the synthesis of major metabolites such as carbon, nitrogen, glycine and glutamine [19]. In Arabidopsis, AtGAPCp1 and AtGAPCp2 may have links to ABA signal transduction and also have a central role in plant primary metabolism and pollen development [20]. Furthermore, the expression of *AtGAPCp1* gene in shoots was induced to varying degrees by cold, osmotic, salinity and drought stresses at early stages [21]. Despite several previous studies, the precise role of *GAPCp* in wheat resistance against abiotic stresses remains unclear. Therefore, it is important to understand the regulatory mechanism of how *TaGAPCps* work in conferring wheat resistance to abiotic stress.

To investigate the relationship between transcriptional levels of *TaGAPCp* genes and abiotic stress tolerance, *TaGAPCp1* gene was cloned and analyzed. Its expression under abiotic stresses was measured by qRT-PCR. In addition, the prey protein Cyt b6f interacting with the TaGAPCp1 protein was screened by Y2H, and was further verified by yeast co-transformation and BiFC analysis. Further experiments demonstrated that the expression of Cyt b6f was significantly elevated under H_2_O_2_ stress. Surprisingly, we found that Cyt b6f has peroxidase activity due to its carotenoids. In conclusion, the results of this study revealed that TaGAPCp1 was indeed involved in the drought stress response of wheat and the stress resistance process of TaGAPCp1 might be accomplished by H_2_O_2_-mediated ABA signaling pathway.

## 2. Results

### 2.1. Identification and Sequence Analysis of TaGAPCp1 Gene in Wheat 

*TaGAPCp1* gene is located on the 6AS chromosome of wheat. The length of *TaGAPCp1* cDNA was 1526 bp, and a complete open reading frame (ORF) of 894 bp encoding a deduced protein of 297 amino acids was obtained. Gene Structure Display Server showed that *TaGAPCp1* contains 9 exons and 8 introns (Figure 1b). The enzyme with a predicted relative molecular mass of 31.54 kDa and an isoelectric point of 6.08 exists as a tetramer of identical subunits (Appendix A), each of which contains two conserved functional domains; an NAD-binding domain (InterPro: IPR0208), and a highly conserved catalytic domain (InterPro: IPR0208) as revealed by the Conserved Domain Database. Appendix A shows that the tertiary structure of TaGAPCp1 is symmetrically distributed, with Alpha helix accounting for 25% and β strand accounting for 30%. Alignment of the amino acid sequence of GAPCps were given in Figure 1a. The results revealed that the amino acid sequence of TaGAPCp1 is shorter than other GAPCps, but the similarity with AtGAPCps was about 80%.

Analyzing the evolutionary relationship of this gene subfamily from different species revealed the evolution of their function and disclosed the potential feature. To further characterize the GAPCp1 protein, the deduced protein and other GAPDHs from *Arabidopsis thaliana* were aligned by Clustalx and MEGA6.0 software via the neighbor-joining algorithm (Appendix A). The alignment results revealed that TaGAPCp1 was very similar to AtGAPCps proteins and belonged to a clade of plant GAPCps based on the previously genome-wide identification and characterization of glyceraldehyde-3-phosphate dehydrogenase genes family in wheat [14].

### 2.2. TaGAPCp1 Responds to Abiotic Stresses

To reveal the inducible expression patterns of *TaGAPCp1* in response to abiotic stresses, qRT-PCR was performed in a Bio-Rad CFX96 system (Bio-Rad Laboratories, Hercules, CA, USA) using the SYBR Green method. According to the qRT-PCR analysis, the expression of *TaGAPCp1* gene in wheat seedling leaf was induced to varying degrees by different chemicals (polyethylene glycol (PEG), NaCl, ABA, 4 °C and H_2_O_2_) at early stages. Exposed to drought stress (PEG), the expression of wheat *TaGAPCp1* was significantly up-regulated, peaking at 6 h. The ABA hormone treatment directly up-regulated expression of *TaGAPCp1*, especially at 6 h. During low temperature and salt treatment, *TaGAPCp1* initially showed slight up-regulated expression and was down-regulated later. However, H_2_O_2_ treatment resulted in up-regulation of *TaGAPCp1* obviously, peaking at 6 h (Figure 2a–e). The expression of *TaGAPCp1* gene in wheat seedling root was basically consistent with leaf and it was generally increased first and then decreased (Figure 2f–j). These results indicate that the expression of *TaGAPCp1* in both leaves and roots was increased under these abiotic stresses relative to the control. However, the up-regulation response of *TaGAPCp1* under the stress of PEG, ABA, and H_2_O_2_ was significant, and the up-regulation response was unapparent under low temperature and salt stress.

To further investigate how intracellular *TaGAPCp1* was maintained in response to drought stress, relative expression of *TaGAPCp1* was determined in stress-treated seedlings. As shown in Figure 3b, the expression of *TaGAPCp1* gene was up-regulated after PEG and ABA treatment alone, but its expression decreased significantly after pretreatment with sodium tungstate solution (Tungstate). However, there was no significant difference between the expression level of *TaGAPCP1* after PEG stress alone and the amount of expression after treatment with dimethylthiourea (DMTU) (Figure 3a). Therefore, it is speculated that ABA may be involved in the regulation of *TaGAPCp1* gene expression under PEG stress.

After treating of OE-*TaGAPCp1* and wild-type (WT) leaves by ABA, it was observed that the stomatal size of both became smaller with the increase of ABA concentration, but it was clear that the OE-*TaGAPCp1* leaf stomata was smaller than WT (Figure 3c). This result demonstrated that TaGAPCp1 plays an important role in ABA-mediated stomata closure. Similarly, we found that the leaf stomata of both of OE-*TaGAPCp1* and WT became smaller as the concentration of H_2_O_2_ increased (Figure 3d). However, the leaf pores of OE-*TaGAPCp1* were smaller relative to WT under the same concentration of H_2_O_2_ treatment. These results revealed that H_2_O_2_ might act as a signal substance in ABA-mediated stomatal closure.

### 2.3. TaGAPCp1 Is Localized to the Plastid

The putative localization of TaGAPCp1 protein was identified by ChloroP prediction server, revealing that it is localized to the plastid. The CDS of *TaGAPCp1* was fused with that of GFP under the control of the Cauliflower *mosaic virus* (35S) promoter to generate a construct expressing a TaGAPCp1-GFP fusion protein (Figure 4a). This fusion vector was used for a more particular knowledge of the subcellular localization of TaGAPCp1. Confocal microscopic examination demonstrated that the TaGAPCp1-GFP fusion vectors were expressed slightly in the plastid of the transformed protoplasts, whereas the pCaMV35S: GFP protein was present throughout the whole cells. These results demonstrated that TaGAPCp1 is probably a plastid protein (Figure 4b).

### 2.4. Screening for the Interacting Proteins of TaGAPCp1

Yeast two-hybrid (Y2H) is a method for identifying interactions between proteins in yeast cells based on transcription factors in the transcriptional regulation of eukaryotic genes [22]. In this study, the wheat cDNA library (prey vector) and bait protein expression plasmid pGBKT7-TaGAPCp1 were constructed, and the full-length of *TaGAPCp1* CDS was fused to GAL4 DNA-binding domain. The expression of the bait vector pGBKT7-TaGAPCp1 in Y2HGold Yeast Strain was verified by colony PCR and agarose gel electrophoresis. Colony PCR of the Y2HGold Yeast Strain transformed with the pGBKT7-TaGAPCp1 plasmid showed bands by gel electrophoresis, whereas the colony PCR of the Y2HGold Yeast Strain transformed with the pGBKT7 plasmid had no band (Figure 5a). It indicated that the pGBKT7-TaGAPCp1 plasmid and the pGBKT7 plasmid were both successfully transformed into Y2HGold Yeast Strain. 

When introduced into yeast Y2HGold Yeast Strain, the absence of blue colonies on the SDO/X plates and no colony growth on the SDO/X/A plates indicated that pGBKT7 and pGBKT7-TaGAPCp1 did not autonomously activate the reporter gene. This indicates that pGBKT7-TaGAPCp1 does not activate MEL1 and AUR1-C reporter gene expression, i.e., glyceraldehyde triphosphate dehydrogenase does not have self-activation. In addition, toxicity tests indicated that the TaGAPCp1 was not toxic to yeast. First, because the size and growth conditions of the colonies were comparable to Y2HGold Yeast Strain transformed with the pGBKT7 empty vector (Figure 5b). Secondly, the concentration of the bacterial liquid containing pGBKT7-TaGAPCp1 was substantially equal to the concentration of the bacterial liquid containing pGBKT7 empty vector (Appendix A). The autoactivation and toxicity test indicated that the constructs were suitable for use in Y2H screening.

The bait strains and wheat cDNA libraries were used for hybrid screenings. After hybridization, the total culture was spotted onto DDO/X/A plates after 4 h incubation. This high stringency screening was repeated at least 2–3 times. There were 186 positive clones identified through screening on double dropout medium DDO/X/A and total of 64 colonies were grown at QDO/X/A solid medium. After two re-identifications on QDO/X/A solid medium, 11 colonies were screened on the defective medium, and all of these colonies grew well and turned blue (Figure 5c). The yeast plasmid PCR results showed that all of the nine selected clones had bands, and the sequences were of different lengths and distributed in the range of 1000–2000 bp (Figure 5d). Sequence (Appendix A) analysis of the plasmids indicated that these 11 plasmids, represent 7 homologous cDNA (Table 1).

### 2.5. Verify the Interaction between TaGAPCp1 and the Interacting Proteins

Each potential positive prey plasmid (pGADT7-X vector) and pGBKT7-TaGAPCp1 was co-transformed into Y2HGold Yeast Strain, respectively. Figure 6a shows that colonies were observed on QDO/X/A plates of P2, P3, P4, and PC, and no colonies were observed on the other plates. To eliminate the false positive result, controls were employed in this experiment. Negative controls for P3 and P4 (Y2HGold Yeast Strain transformed with pGBKT7 empty bait plasmid and pGADT7-X vector) showed colonies on QDO/X/A plates, while no colonies were observed on QDO/X/A plate for P2 negative control. On the other hand, blue colonies of the positive control (Y2HGold Yeast Strain containing pGADT7-T and pGBKT7-53) were observed on the QDO/X/A plate. Therefore, there is an interaction relationship between P2 and TaGAPCp1. P2 was identified as Cytochrome b6-f Complex Iron-sulfur Subunit (Cyt b6f) protein (GenBank: XM_020345741.1) by DNA sequencing. Further confirming the interaction between Cyt b6f and TaGAPCp1 in wheat cells, pSPYNE-TaGAPCp1 (yellow fluorescent N-terminal fragment protein) and pSPYCE-Cyt b6f (C-terminal fragment of a yellow fluorescent protein) recombinant vector were transformed to tobacco and prepared protoplasts; the fluorescence signal was mainly observed in the cytoplasm. In contrast, no positive signals were observed when pSPYNE-TaGAPCp1 and pSPYCE as well as pSPYNE and pSPYCE-Cyt b6f constructs were co-transformed into tobacco protoplasts (Figure 6b).

### 2.6. Analysis of the Interacting Protein Cytb6f

The Cytb6f gene ORF sequence is 669 bp and encodes 222 amino acids. In addition, the analysis results of ProtParam software showed that the Molecular weight of the protein was 23.7 kD, the Theoretical pI was 8.47 and the instability index (II) was computed to be 25.34, classifying the protein as stable. Moreover, the average hydrophilicity index Grand average of hydropathicity (GRAVY) was −0.071 and this identified the protein as hydrophilic. There is considerable evidence to suggest that Cytb6f may exist as a dimer, with each monomer composed of eight subunits distributed on the thylakoid membrane [23]. In plants, these thylakoid membrane protein complex subunits are encoded by chloroplasts and nucleus. To investigate the activity of Cyt b6f under abiotic stress, the effects of PEG, ABA, and H_2_O_2_ on wheat leaves were examined. As shown in Figure 7b, the expression level of Cyt b6f showed a significant increase under H_2_O_2_ stress and reached a peak at 2 h. The results indicate that H_2_O_2_ was first proposed as a signaling molecule involved in the process of Cyt b6f response to abiotic stress. Studies also have reported that Cyt b6f monomer molecules contain one molecule of natural component chlorophyll a (Chl a) and one molecule carotenoid [24,25]. Additionally, the chlorophyll content in OE-*TaGAPCp1* plants was significantly higher than that in WT (Figure 7a).

## 3. Discussion

### 3.1. TaGAPCp1 Plays a Key Role in Abiotic Stresses

Plastidial GAPCps play an important physiological role in plant function [26]. It has been reported that TaGAPCp1 is involved in plastidial glycolytic pathway and plays a specific role in glycolytic energy production in non-green plastids and chloroplasts [6]. In this study, the TaGAPCp1 was identified and analyzed from Chinese Spring wheat (Figure 1). The gene shares high similarity with GAPCps members of other species, which further illustrate that the *GAPDH* coding region is highly conserved [27].

The substrate of GAPCp, glyceraldehyde-3-phosphate, is the first precursor of the methyl-erythritol phosphate pathway responsible for ABA biosynthesis in the plastids [5,6]. In the present study, evidence showed that TaGAPCp1 is a positive regulator of wheat tolerance to drought, ABA and H_2_O_2_ stress. The relative expression levels of *TaGAPCp1* in plants under drought, ABA and H_2_O_2_ stress were significantly higher than those in non-stressed plants (Figure 2). The same results indicated that the enzyme activity of GAPDH in *Craterostigma plantagineum* substantially increased under ABA treatment [28]. These protective responses of plants might be primarily regulated by synergistic and antagonistic actions of signaling molecules. In any case, these results indicate that TaGAPCp1 is an important player in the response of wheat to drought, ABA and H_2_O_2_ stress.

In addition, plant ABA plays an important role in plant responses to abiotic stress [29]. Additionally, plants rapidly inhibit stomatal opening under the control of ABA signaling pathway, in order to preserve water [30]. ABA hormones were reported to have a close connection with the signaling networks and the regulation of developmental processes responding to variously environmental stresses. ABA plays an important role in plant responses to abiotic stress [29]. It has been shown that the expression level of *TaGAPCp1* was relatively high under the stress of PEG, although the expression level was significantly decreased after pretreatment with ABA inhibitor (Figure 3b). Thus, it could be speculated that drought stress may cause changes in the osmotic pressure of plant cells and then pass signals through the “osmotic sensing device” to the cells. Whereafter, a series of enzymes in the cell regulate the expression of *TaGAPCp1* gene under drought stress through the ABA signaling pathway. Additionally, plants rapidly inhibit stomatal opening under the control of ABA signaling pathway (Figure 3c), in order to preserve water [30]. However, the leaf stomata of both of OE-*TaGAPCp1* and WT became smaller after H_2_O_2_ treatment (Figure 3d). Therefore, TaGAPCp1 plays an important role in responding to water-deficient abiotic stress and it is likely to be regulated by the H_2_O_2_-dependent ABA signaling pathway.

### 3.2. The Interaction Protein Cyt b6f of TaGAPCp1 Has Peroxidase Activity

The interaction protein Cyt b6f of TaGAPCp1 was obtained by yeast two-hybrid system (Figure 5) and verified by BiFC method (Figure 6b). This further illustrated the strong interaction between Cyt b6f and TaGAPCp1. Cyt b6f occupies a central position in the photosynthetic electron transport chain, bridging the photosynthetic reaction centers of PS I and PS II [31]. It may exist as a dimer, with each monomer composed of eight subunits [23]. These subunits evenly distribute on the thylakoid membrane. In plants, these thylakoid membrane protein complex subunits are encoded by chloroplasts and nucleus [28]. In addition, two cofactors, Chl a and carotenoids, have been identified in a 1:1:1 ratio with the monomeric Cyt b6f. It is worth mentioning that Cyt b6f is also a heme-containing protein complex with peroxidase activity [24,25]. 

### 3.3. The Role of TaGAPCp1 in Drought Resistance Is Largely Dependent on H_2_O_2_-Mediated ABA Signaling Pathway

On a molecular level, reactive oxygen species (ROS) was a severe problem of abiotic stress in plants [32]. Excessive ROS production could cause oxidative damage to the photosynthetic apparatus, block electron transport and decrease photosynthetic capacity [33]. More previous studies indicate that ROS would accumulate in stressful environments, such as drought, salinity, ABA, and other conditions [34,35,36]. It has been reported that carotenoids are important powerful antioxidants [37]. Because of the powerful antioxidant activity, carotenoids maintain ROS homeostasis and stabilize cellular membranes, thus improving plant abiotic stress tolerance [32,38,39]. In this study, the relative expression of Cyt b6f in wheat leaves was significantly increased under H_2_O_2_ stress (Figure 7b). We also found that TaGAPCp1 promotes the synthesis of chlorophyll a and carotenoids, which are the main component of Cyt b6f (Figure 7a). Furthermore, antioxidant carotenoid, the important constituents of Cyt b6f, may reduce ROS accumulation by inhibiting H_2_O_2_ production [32]. Carotenoids are also important precursors of phytohormones, including ABA [40]. Moreover, ABA can induce up-regulation of antioxidants such as SOD, CAT and carotenoids, thereby increasing the ability of plants to scavenge ROS [41]. Thus, ROS may participate in ABA signal transduction [42,43,44,45]. H_2_O_2_ is the main form of reactive oxygen species (ROS), which acts as a crucial signaling molecule in the ABA response pathway in guard cells [46,47]. Additionally, Cyt b6f is a major component of cyclic electron transport chain around PS I which plays an important role in dissipating excessive excitation energy [33]. Carotenoids play essential roles in various light-harvesting processes in plants and help protect the photosynthetic machinery from photo-oxidative damage. There are reports that the *orange* gene of sweet potato helps maintain carotenoid homeostasis and directly stabilize PSII to improve plant tolerance to environmental stress [39]. Therefore, we could speculate that Cyt b6f might play an important role in TaGAPCp1 response to drought and other stress processes through ABA signaling pathway (Figure 8). However, the activity of TaGAPCp1 and the interaction protein Cyt b6f were significantly increased under the stress of drought and H_2_O_2_ stress, respectively (Figure 8). It is also fully demonstrated that the role of TaGAPCp1 in drought resistance is largely determined by H_2_O_2_-mediated ABA signaling pathway.

## 4. Materials and Methods

### 4.1. Plant Materials and Abiotic Stresses

Wheat seeds (Chinese Spring) were sterilized with 0.01% HgCl_2_ and washed at least three times with distilled water. Sterilized seeds were grown hydroponically in glass dishes in full-strength Hoagland solution. Germinated seedlings were maintained under a relative humidity (RH) of 75% and 16/8 h (light/dark, 25/15 °C) photoperiod (light intensity of 250 µmol m^−2^s^−1^) for 14 days. The two-leaf-stage seedlings were immersed in NaCl (250 mM), water deficiency (20% PEG 8000), H_2_O_2_ (10 mM), cold (4 °C) and ABA (100 µM abscisic acid), respectively. All of the treatments and controls were sampled at 0, 2, 4, 6, 8, 12 and 24 h with three biological replicates, each of which composed of 6 wheat seedlings.

Furthermore, wheat seedlings were treated with 1 mM sodium tungstate solution (Tungstate) for 6 h and prior to further treatment with PEG 8000 for more 6 h. At the same time, seedlings treated with 20% PEG 8000, 100 μM abscisic acid and 1 mM sodium tungstate for 6 h, respectively, were used as controls. The seedlings were also treated with 5 mM DMTU for 6 h, then treated with PEG 8000 for 6 h; while those treated with 20% PEG8000, 10 mM H_2_O_2_ and 5 mM DMTU for 6 h, respectively, were used as a control. Tungstate was used as an ABA inhibitor and DMTU as an H_2_O_2_ inhibitor. The samples were immediately frozen in liquid nitrogen and stored at −80 °C for extraction of total RNA.

Arabidopsis seeds were germinated on 1/2 Murashige and Skoog (MS) medium containing 2% sucrose and then transplanted into soil. The plants were grown at 16/8 h (light/dark, 25/15 °C) photoperiod (light intensity of 250 µmol m^−2^s^−1^). To construct a *TaGAPCp1* overexpressing plant (OE-*TaGAPCp1* plants), the coding region of *TaGAPCp1* was introduced into the plant transformation vector pCAMBIA1302 under the control of the 35S promoter. The construct was transformed into Agrobacterium and transformed into wild type Arabidopsis by invasive inflorescence. 

### 4.2. Identification and Sequence Analysis of TaGAPCp1

The coding sequence of *TaGAPCp1* (MF477938.1) was amplified from cDNA extracted from Chinese spring wheat leaves using primers TaGAPCp1-F and TaGAPCp1-R (Appendix A) based on the nucleotide sequence of *TaGAPCp1* and the amplified sequence was blasted against the Triticum aestivum databases (http://plants.ensembl.org/index.html). The exon-intron structures of gene were identified through the online Gene Structure Display Serve (http://gsds.cbi.pku.edu.cn). To analyze the TaGAPCp1 protein well, the physical and chemical properties of the protein were determined on the ProtParam (https://web.expasy.org/protparam/) and the sequence of amino acid was compared by DNAMAN6.0 (Lynnon Biosoft, San Ramon, CA, USA). In addition, the tertiary structure of TaGAPCp1 was predicted by SWISS-MODEL (http://swissmodel.expasy.org/). The sequence was determined using NCBI database (http://www.ncbi.nlm.nih.gov/) and domains of the proteins were analyzed by the InterProScan (http://www.ebi.ac.uk/cgi-bin/iprscan/). Subsequently, a phylogenetic tree between species was constructed using the Neighbor-joining method in MEGA 6.0 software (Arizona State University, USA).

### 4.3. Transcript Level of TaGAPCp1

Total RNA was extracted from the wheat seedling by RNAios reagent kit (TaKaRa, Kusatsu, Japan) and the first strand of cDNA was synthesized using the PrimeScript™ RT reagent kit (TaKaRa) according to the manufacturer’s instructions. For the qRT-PCR analyses, the wheat β-actin was used as the internal reference. Quantitative real-time PCR was performed in a Bio-Rad CFX96 system (Bio-Rad Laboratories) using SYBR Premix Ex Taq (TaKaRa). The used primers are listed in Appendix A. The volume of the PCR reaction was 20 μL under the following conditions: 95 °C for 30 s, 40 cycles of 95 °C for 5 s, 60 °C for 30 s, and melting curve analysis. The qRT-PCR analysis was accomplished with comparative 2^−ΔΔ*C*t^ method [48] by Bio Rad CFX Manager software and SPSS 20. Then the relative expression figure of TaGAPCp1 were showed via using Origin 8.0 (Microcal, CA, USA).

### 4.4. Stomatal Aperture Measurement

Mature leaves of 4-week-old Arabidopsis plants were immersed in stomatal opening solution (30 mM KCl, 100 mM CaCl_2_, and 10 mM MES, pH 6.15) for 2 h at 22 °C [49]. The treated leaves were transferred to treatment solutions containing different concentrations of ABA (0, 10, and 100 μM) and H_2_O_2_ (0, 100, and 200 μM). After incubating for 2 h under white light, the stomata of each exfoliated epidermis were observed [50].

### 4.5. Subcellular Localization of TaGAPCp1

To identify the subcellular localization of TaGAPCp1, the specific primers 1302F/R (Appendix A) without the terminating codon were used to amplify the *TaGAPCp1* coding sequence (CDS). PCR products were inserted into the corresponding sites of pCMBIA1302-GFP vector (digested with NcoI) with One Step Cloning Kit (Vazyme, Nanjing, China). The pCaMV35S:TaGAPCp1-GFP fusion vector was used for *Agrobacterium*-mediated transient transformation of 4-week tobacco epidermal cells and pCMBIA1302-GFP vector used as the control. Protoplasts isolated from tobacco epidermal cells maintain many of the same physiological responses and cellular activities as intact plants [51,52] GFP signals were photographed using an Andor Confocal Laser Scanning Microscopy (LSM710, Karl Zeiss, Jena, Germany).

### 4.6. Screening of Proteins Interacting with TaGAPCp1 by Yeast Two-Hybrid Assay

Yeast biology was directly used to construct a Mate & Plate library via in vivo recombination between Chinese spring wheat cDNA and the Matchmaker prey vector pGADT7-Rec. The CDS of *TaGAPCp1* was cloned into the pGBKT7 vector by primers TaGAPCp1-F/R (Appendix A) and transformed into the yeast strain Y2H Gold according to the manufacturer’s instructions (Clontech, Mountain View, CA, USA). It was important to test the bait protein for auto-activation and toxicity prior to Y2H screening. Thus, the pGBKT7-TaGAPCp1 and pGBKT7 empty vector were transformed into Y2HGold Yeast Strain, respectively. Subsequently, the cultures were separately plated on SD/-Trp (SDO), SD/-Trp/X-α-Gal (SDO/X) and SD/-Trp/X-α-Gal/AbA (SDO/X/A) agar plates and incubated at 30 °C for 5 days. Among them, the final concentrations of X-α-Gal and AbA were 40 μg/mL and 200 ng/mL, respectively. A comparison with the colonies transformed with pGBKT7 empty vector made it easy to distinguish if the bait plasmid had induced auto-activation and toxicity. Y2H system was performed between recombinant pGBKT7-TaGAPCp1 (bait) and wheat cDNA library (prey) (Clontech). The bait strain was co-cultured with the library strain Y187 until the fertilized egg was examined under a microscope (40×). To screen the cDNA library, the mated cells were coated on the 150-mm double dropout medium SD/-Leu/-Trp/X-α-Gal/AbA (DDO/X/A) and SD/-Ade/-His/-Leu/-Trp/X-α-Gal/AbA (QDO/X/A).

### 4.7. Analysis of the Positive Preys

When the interaction was confirmed as positive, the insert sequence of the prey plasmid was sequenced by the T7 sequencing primer. We ensured that the open reading frame (ORF) was fused in frame to the GAL4 transcriptional activation domain. The insert sequence of the positive preys was subjected to BLAST search (https://blast.ncbi.nlm.nih.gov/Blast.cgi) to identify the corresponding *Triticum aestivum* genes.

### 4.8. Identification of Prey Proteins Interacting with TaGAPCp1

Prey proteins were subcloned into the pGADT7 prey vector. The interaction of preys with TaGAPCp1 was confirmed by co-transforming into Y2HGold Yeast Strain with the bait (TaGAPCp1 in pGBKT7) vector. In addition, the pGBKT7-p53 and pGBKT7-Lam bait plasmids were separately transformed into Y2HGold Yeast Strain with the prey plasmid (pGADT7-T) as positive and negative controls. Moreover, the interaction of Cyt b6f with TaGAPCp1 was also verified by BiFC method and the used primers were listed in Appendix A. The transient assay was performed by protoplasts cells of tobacco leaves, and yellow fluorescent protein (YFP) fluorescence was observed with confocal laser scanning microscopy (LSM710, Karl Zeiss). YFP fluorescence signals were collected in the 500–570 nm wavelength range. For chloroplast autofluorescence, the wavelength range monitored was 630–700 nm [53,54].

### 4.9. Identification and Sequence Analysis of Cyt b6f

The physical and chemical properties of the protein were determined on the ProtParam (https://web.expasy.org/protparam/). To analyze the Cyt b6f protein well, its expression under abiotic stresses was measured by qRT-PCR. In addition, to analyze the relationship between Cytb6f and TaGAPCp1, we measured the chlorophyll content of OE-*TaGAPCp1* and wild type Arabidopsis, respectively (Acetone method).

## 5. Conclusions

In conclusion, our data suggested that TaGAPCp1 plays an important role in responding to drought stress by the ABA signaling pathway. At the same time, we revealed that H_2_O_2_ could be used as a signaling molecule in the process of Cyt b6f response to abiotic stress. Since the interaction between Cyt b6f and TaGAPCp1 was confirmed, it was speculated that the stress resistance process of TaGAPCp1 might probably be completed by H_2_O_2_-mediated ABA signaling pathway with H_2_O_2_ acting as a signal molecule, while the antioxidant activity of carotenoids in Cyt b6f could probably maintain the relative balance of ROS. These findings reveal TaGAPCp1 is critical for abiotic stress responses in wheat.

## Figures and Tables

**Figure 1 ijms-20-01104-f001:**
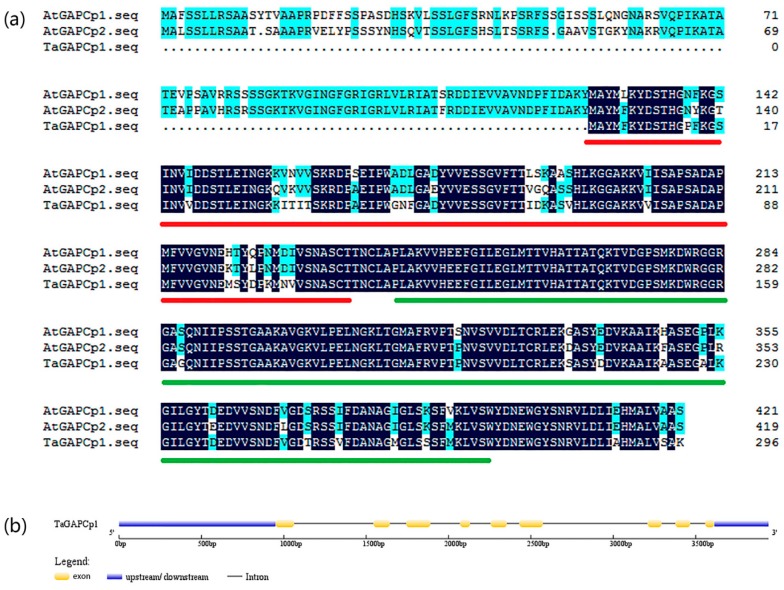
Sequence analysis. (**a**) Alignment of the Amino acid sequence of *TaGAPCps*. The identical and 75% amino acid sequence similarity are separately indicated by mazarine and green color. Red underlined, Glyceraldehyde 3-phosphate dehydrogenase, NAD(P) binding domain (IPR0208); green underlined, Glyceraldehyde 3-phosphate dehydrogenase, catalytic domain (IPR0208). (**b**) Schematic diagram for exons/introns and upstream/downstream structures of *TaGAPCp1*. Exons, introns, upstream/downstream are indicated by yellow boxes, black horizontal lines, and blue boxes, respectively.

**Figure 2 ijms-20-01104-f002:**
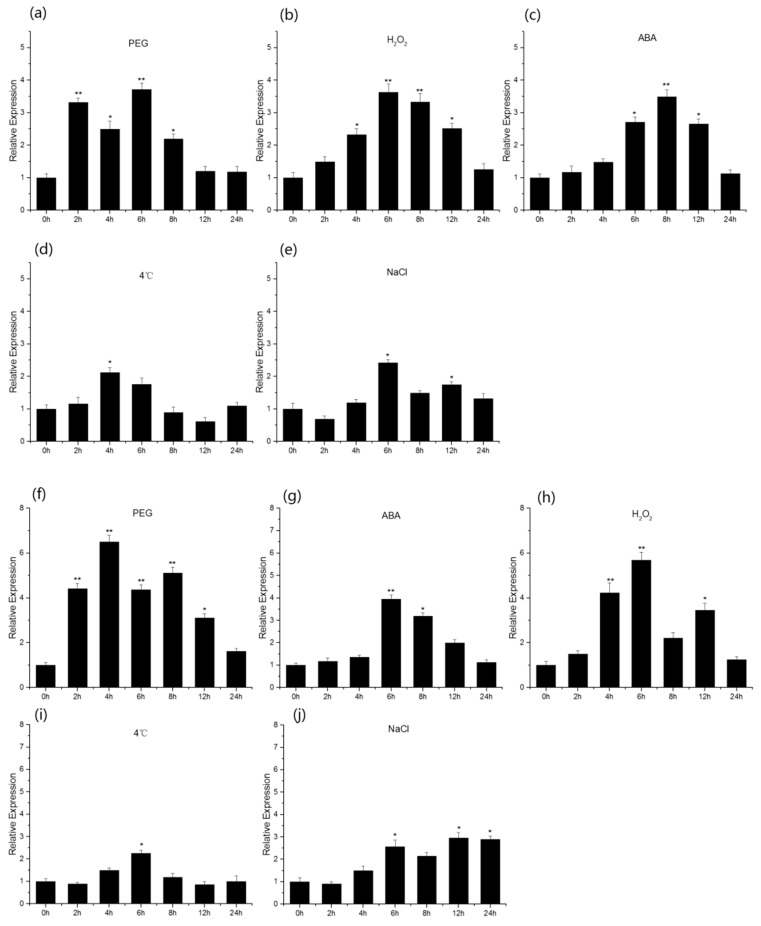
Expression profiles of TaGAPCp1 in wheat. (**a**–**e**) expression patterns of TaGAPCp1 in wheat leaves in response to various abiotic stresses: PEG, ABA (abscisic acid), H_2_O_2_, 4 °C, NaCl. (**f**–**j**) Expression patterns of TaGAPCp1 in wheat roots in response to various abiotic stresses: PEG, ABA (abscisic acid), H_2_O_2_, 4 °C, NaCl. Significant differences were detected by SPSS analysis. * *p* < 0.05, ** *p* ≤ 0.01.

**Figure 3 ijms-20-01104-f003:**
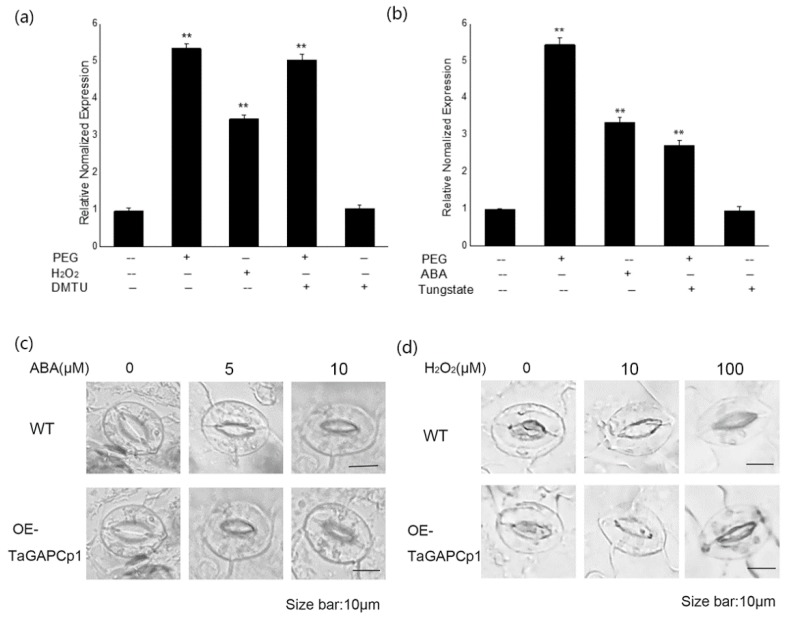
Effects of ABA and H_2_O_2_ inhibitors on the expression of TaGAPCp1 in wheat and the effects of ABA and H_2_O_2_ on the stomatal of transgenic arabidopsis leaves. (**a**) expression of aGAPCp1 gene in wheat seedling leaves under PEG stress after pretreatment with H_2_O_2_ inhibitor (dimethylthiourea (DMTU)). (**b**) expression of *TaGAPCp1* gene in wheat seedling leaves under PEG stress after ABA inhibitor (tungstate) pretreatment. Bars are shown as the mean ± Standard Deviation (SD). Significant differences were detected by SPSS analysis. * *p* < 0.05, ** *p* ≤ 0.01. (**c**,**d**) Stomatal size of WT and OE-TaGAPCp1 plants. Stomatal closure was observed after incubation of mature leaves in ABA (0, 5 and 10 μM) and H_2_O_2_ (0, 10, and 100 μM) buffers for 2 h, respectively. Each experiment was performed in triplicate. Bar = 10 μm.

**Figure 4 ijms-20-01104-f004:**
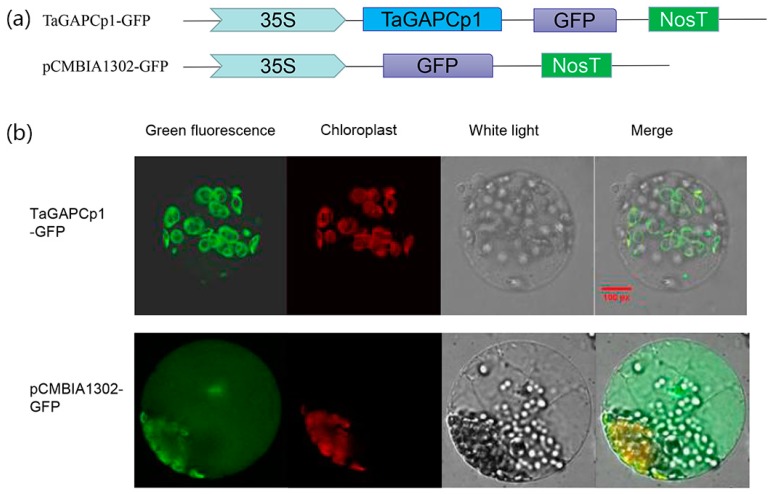
Subcellular localization of TaGAPCp1 in tobacco cells. (**a**) Structure of TaGAPCp1-GFP fusion expression vector. (**b**) The images showed the expression of the pCaMV35s: TaGAPCp1-GFP fusion protein in tobacco cells. All of the images were obtained using a confocal microscope. Bar = 20 μm.

**Figure 5 ijms-20-01104-f005:**
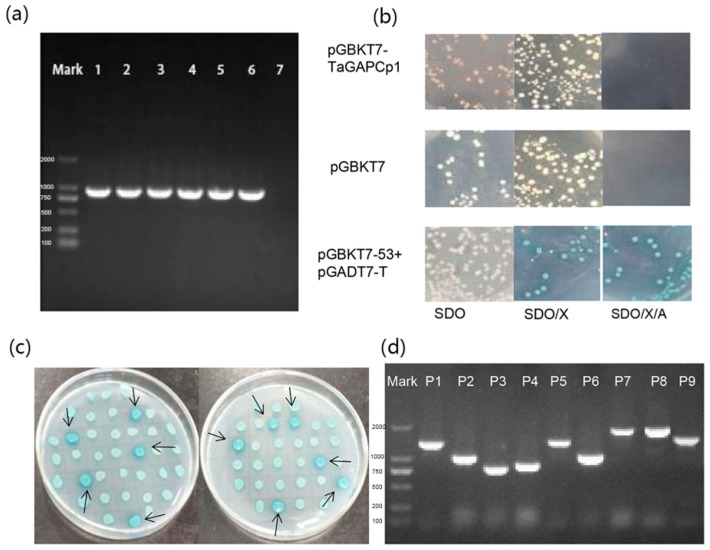
Screening of yeast two-hybrid. (**a**) Gel electrophoresis analysis of colony PCR about the Y2HGold Yeast Strain transformed with pGBKT7-TaGAPCp1 plasmids (number 1–6) and pGBKT7 plasmids (number 7). Mark is DL 2000 molecular marker and from top to bottom are 2000, 1000, 750, 500, 200 and 100. (**b**) Determination of autoactivation and toxicity of the bait vector in Y2HGold Yeast Strain on different selection mediums. The first line was the result of transforming pGBKT7-TaGAPCp1 into Y2HGold Yeast Strain alone; the second line (converting pGBKT7 to Y2HGold Yeast Strain) was used as a negative control; the third line was to convert pGBKT7-53 and pGADT7-T into Y2HGold Yeast Strain as positive control. (**c**) Screening diagram of TaGAPCp1. Blue clones indicated positive results, whereas white or absent clones were negative; (**d**) PCR amplification of positive prey plasmids by T7 primers. Lane P1–P9: PCR products amplified from the positives P1–P9 (responding to prey vectors P1–P9); Lane Mark: DL 2000 marker.

**Figure 6 ijms-20-01104-f006:**
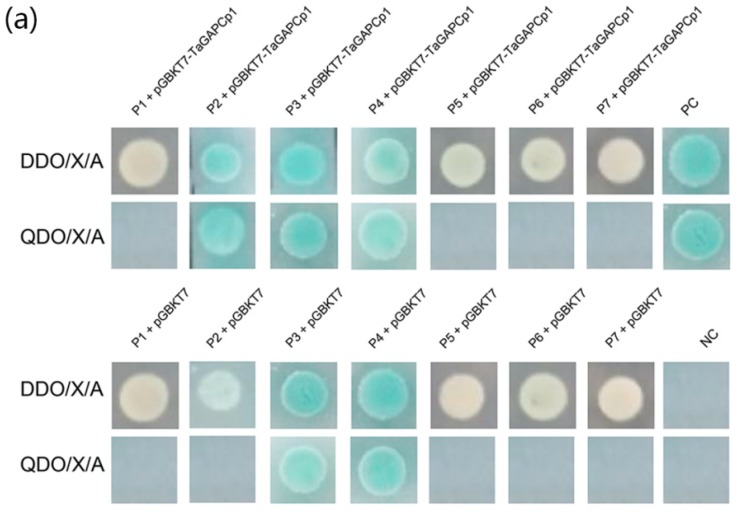
Identification of prey proteins interacting with TaGAPCp1. (**a**) Confirmation of true positive clones by small-scale Y2H assay. First row and second row: pGBKT7-TaGAPCp1 plasmid and the respective prey 1 to prey 7 plasmids (P1–P7) was co-transformed into Y2HGold Yeast Strain then staked on QDO/X/A plates and DDO/X/A plates, respectively. Third row and fourth row: pGBKT7-TaGAPCp1 plasmid and pGBKT7 empty bait was co-transformed into Y2HGold Yeast Strain then staked on QDO/X/A plates and DDO/X/A plates, respectively. PC indicates a positive control, NC indicates a negative control. (**b**) BiFC assay of the interaction between TaGAPCp1 and Cyt b6f proteins in tobacco leaf protoplasts. The pSPYNE-TaGAPCp1 and pSPYCE-Cyt b6f constructs were co-infiltrated in tobacco by Agrobacterium. The YFP fluorescence was detected by confocal laser scanning microscopy. Co-transformants of pSPYNE-TaGAPCp1 and pSPYCE as well as pSPYNE and pSPYCE-Cyt b6f were used as negative controls. Bar = 20 μm.

**Figure 7 ijms-20-01104-f007:**
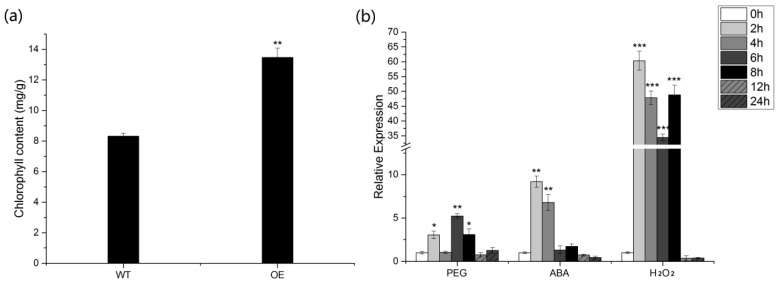
Chlorophyll content in Arabidopsis plants and Expression patterns of Cyt b6f in wheat leave. (**a**) Chlorophyll content in OE-*TaGAPCp1* plants and wild-type plants. OE represents OE-*TaGAPCp1* plants; WT represents wild-type plants. (**b**) Expression patterns of Cyt b6f in wheat leaves in response to various abiotic stresses: PEG, ABA and H_2_O_2_. Bars are shown as the mean ± SD. Significant differences were detected by SPSS analysis. * *p* < 0.05; ** *p* ≤ 0.01; *** *p* ≤ 0.001.

**Figure 8 ijms-20-01104-f008:**
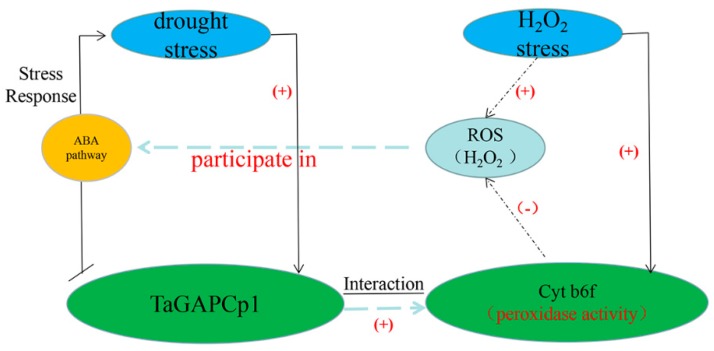
Model of how TaGAPCp1 responds to drought stress. Drought stress and H_2_O_2_ stress significantly increased the expression levels of TaGAPCp1 and Cyt b6f, respectively. TaGAPCp1 plays an active role in response to drought stress by H_2_O_2−_dependent ABA signaling pathway. The interaction between TaGAPCp1 and Cyt b6f has been confirmed. TaGAPCp1 also promotes the biosynthesis of chlorophyll, which is a natural component of Cyt b6f. Cyt b6f can reduce the accumulation of reactive oxygen species (ROS) and then respond to abiotic threats. Among them, (+) represents promotion and (−) represents inhibition. Straight lines represent direct effects and dashed lines represent indirect effects.

**Table 1 ijms-20-01104-t001:** Basic Local Alignment Search Tool (BLAST) result for interacting proteins between TaGAPCp1 and cDNA wheat library.

Clone	Name	Functions	GenBank
1,5	Tauschii dormancy-associated protein homolog, mRNA	Involved in plant stress resistance	XM_020318726.1
2	Tauschii cytochrome b6-f complex iron-sulfur subunit, mRNA (Cyt b6f)	Involved in photoreactive electron transport	XM_020345741.1
3,11	Tauschii ATP-dependent Clp protease proteolytic subunit 6, mRNA	Involved in Stress response	XM_020331538.1
4,8,10	Tauschii ribulose-1,5-bisphosphate carboxylase, complete cds	Involved in Calvin cycle	KT288199.1
6	Tauschii GATA transcription factor, mRNA	Transcription factor with a special zinc finger structure	XM_020324318.1
7	Tauschii probable E3 ubiquitin-protein ligase, mRNA	One of the key enzymes in the ubiquitination process	XM_020328679.1
9	Tauschii protein SGT1, mRNA	Participate in the regulation of plant disease resistance	KJ907387.1

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
