# Peer review of "The Plastidial Glyceraldehyde-3-Phosphate Dehydrogenase Is Critical for Abiotic Stress Response in Wheat"

_ijms, 2019, doi:10.3390/ijms20051104_

Round 1
Reviewer 1 Report
Please write the full name of the organism you studied? Wheat (Triticum aestivum L.)?
Chinese Spring can be regarded as a drought-resistant variety but I suggest using of the word “Response” instead of “Resistance” in the title.
In page 6, line 157, how does mCherry represent chloroplast autofluorescence?
In page 14, line 435, you say that “In addition, to analyze the relationship between Cytb6f and TaGAPCp1, we measured the chlorophyll content of OE-TaGAPCp1 and wild type Arabidopsis, respectively (Acetone method).” How do you explain the relationship between Cytb6f and TaGAPCp1 in terms of chlorophyll content?
In page 11, line 278, you say that “However, the leaf stomata of both of OE-TaGAPCp1 and WT became smaller after H2O2 treatment”. H2O2 is known to be involved in ABA signaling and therefore closure of stomata after H2O2 treatment is not surprising.
Remove experimental details from discussion section.
In legend of fig 8, you show that ABA represses TaGAPCp1. Is it correct? However your data (fig 2) suggest that ABA treatment induces expression of TaGAPCp1.
I also highlighted a few stylistic remarks on the manuscript (in attachment).

Author Response
Dear reviewer,
Thank you for your letter and your comments concerning our entitled “The Plastidial Glyceraldehyde-3-Phosphate Dehydrogenase Is Critical for Abiotic Stress Response in Wheat”. Those comments are all valuable and very helpful for revising and improving our paper, as well as the important guiding significance to our researches. We have studied comments carefully and have made correction which we hope meet with approval. The responds to the reviewer’s comments are as following:
Point 1: Please write the full name of the organism you studied? Wheat (Triticum aestivum L.)?
Response 1: The full name of the organism we studied is Triticum aestivum L and has been written in the article.
Point 2: Chinese Spring can be regarded as a drought-resistant variety but I suggest using of the word “Response” instead of “Resistance” in the title.
Response 2: Based on your comments and suggestions, we carefully considered the title of the manuscript and we have changed the word “Resistance” into “Response”.
Point 3: In page 6, line 157, how does mCherry represent chloroplast autofluorescence?
Response 3: mCherry is a kind of red fluorescent protein, which has the advantages of good fluorescence intensity and light stability. Its maximum excitation and emission are 587 and 610 nm, respectively. In our manuscript, our intention is to show the red light of the chloroplast autofluorescence, but neglect that mCherry itself is a red fluorescent protein. After your suggestion, we think it is not appropriate to use “mCherry”, we will use “Chloroplast” to label the picture.
Point 4: In page 14, line 435, you say that “In addition, to analyze the relationship between Cytb6f and TaGAPCp1, we measured the chlorophyll content of OE-TaGAPCp1 and wild type Arabidopsis, respectively (Acetone method).” How do you explain the relationship between Cytb6f and TaGAPCp1 in terms of chlorophyll content?
Response 4: According to the relevant literature, each molecule of Cyt b6f contains 1 molecule of Chlorophyll a (Chl a). By measuring the chlorophyll content OE-TaGAPCp1 and wild type Arabidopsis, we can analyze the the relationship between Cytb6f and TaGAPCp1 and draw a conclusion that they are positively related.
Point 5: In page 11, line 278, you say that “However, the leaf stomata of both of OE-TaGAPCp1 and WT became smaller after H2O2 treatment”. H2O2 is known to be involved in ABA signaling and therefore closure of stomata after H2O2 treatment is not surprising.
Response 5: There is no doubt that we can understand your thought. However, we aim to compare the difference in stomatal opening degree between wild type and OE-TaGAPCp1 arabidopsis after H2O2 treatment. From our experiments, we can conclude that the leaf stomata of OE-TaGAPCp1 is smaller than wild type arabidopsis.
Point 6: Remove experimental details from discussion section.
Response 6: Thank you for your valuable comments, we have already removed experimental details from discussion section.
Point 7: In legend of fig 8, you show that ABA represses TaGAPCp1. Is it correct? However your data (fig 2) suggest that ABA treatment induces expression of TaGAPCp1.
Response 7: In the pattern diagram of Fig 8, we explain how TaGAPCp1 responds to drought stress. Drought stress and H2O2 stress significantly increased the expression levels of TaGAPCp1 and Cyt b6f, respectively. In addition, ABA treatment induces expression of TaGAPCp1. TaGAPCp1 plays an active role in response to drought stress by H2O2-dependent ABA signaling pathway. The interaction between TaGAPCp1 and Cyt b6f has been confirmed. TaGAPCp1 also promotes the biosynthesis of chlorophyll, which is a natural component of Cyt b6f. Cyt b6f can reduce the accumulation of ROS (H2O2) and then respond to abiotic threats. Figure 8 does not show that ABA inhibits TaGAPCp1.
Point 8: I also highlighted a few stylistic remarks on the manuscript (in attachment).
Response 8: Thank you very much for reviewing our article very seriously and for making a lot of valuable comments. I have modified it according to your remarks, and these opinions have helped me a lot.
Point 9: The main revision of the paper can be seen in the attachment.
Reviewer 2 Report
The authors have been tried to show the role of TaGAPCp protein as one positive regulator in against to abiotic stresses.
- Please find the attachment and follow the comments.
-In addition,
1- Line 283-287 should be transferred in material and Methods.
2- The authors have been referred to the 53 references. 19 references of these are old to refer and should be up-to-date published in 2010 or above. According to the policy of journal, at least 85% of references should be up-to-date.

Author Response
Dear Reviewer, Thank you for your letter and for your comments concerning our entitled “The Plastidial Glyceraldehyde-3-Phosphate Dehydrogenase Is Critical for Abiotic Stress Response in Wheat”. Those comments are all valuable and very helpful for revising and improving our paper, as well as the important guiding significance to our researches. We have studied comments carefully and have made correction which we hope meet with approval. The responds to the reviewer’s comments are as following:
Point 1:Line 283-287 should be transferred in material and methods.
Response1:Thank you very much for your valuable comments. We have already transferred Line 283-287 in material and methods.
Point 2:The authors have been referred to the 53 references. 19 references of these are old to refer and should be up-to-date published in 2010 or above. According to the policy of journal, at least 85% of references should be up-to-date.
Response 2:Thank you very much for your valuable advice.It is true that some of the references in our article are old, but some of them are more authoritative.In addition, we will update some relatively new references according to the policy of the magazine.
Point 3: Figure S1b: Taking a look at this figure Alpha helix seems occupied more percentage than Beta helix strands (30%). Is that right?
Response 3:Because the structure of the protein is three-dimensional, there will be different results from different perspectives. The following figure shows that another perspective after adjustment. From the picture we can see the clear result, ie Beta helix strands occupied more percentage than Alpha helix. This also matches our predictions.
Point 4: In figure 1b, better use exon instead of CDS when showing the genomic DNA.
Response 4:Thank you very much for your comment , we have changed CDS into exon.
Point 5: Line 101. The authors resulted that TaGAPCp1 was significantly up-regulated by PEG at 2h, while regarding to the fig.2a-e it was shown at 6h. In addition, it has been mentioned up regulated by ABA at 2h, while related figure showed at 8h. Furthermore, (line 105) the authors have been resulted that TaGAPCp1 expression in roots was basically consistent with leaf. In leaf it has been shown a down regulation of……
Response 5:Thank you very much for your question. I am sorry that the numerical value in the article is wrong due to our negligence. We have corrected the mistakes in the article. Thank you again for your valuable questions.
Point 6: Line 120: Why the authors resulted that there is no significant differences between PEG and H2O2 (alone)(fig.3a), while regarding to the fig.3b, there is significant differences between PEG and ABA (alone) with the same relative normalized expression?
Response 6:Thank you very much for your question. The expression of TaGAPCp1 gene was up-regulated after PEG and ABA treatment alone, but its expression decreased significant after pretreatment with sodium tungstate solution (Tungstate) (Fig. 3b). However, there was no significant difference between the expression level of TaGAPCP1 after PEG stressa lone and the Amount of expression after treatment with dimethylthiourea (DMTU) (Fig. 3a).
Point 7: Please provide one schematic figure (BiFC) to add manuscript.
Response 7:The schematic figure (BiFC) has been added to manuscript.
Point 8: Please add this documents (Cyt b6f protein) as supplementary data.
Response 8:The DNA sequence encoding the Cyt b6f protein has been added to supplementary data.
Point 9: Please confirm are these analyses right? Beacause regarding to between H2O2 results and 0h should be *** with a P<0.001.< span="">
Response 9: Thank you very much for your question. Now, let me make a couple of points. First, we are sure that our analysis process is correct, and we usually use "*, P<0.05; **, P < 0.01". In addition, the P value analyzed in figure 7b is indeed less than 0.001. We will take your opinion and re-analyze the data.
The main revision in the paper can be seen in the attachment